# InstaTAP: Instance Motion Estimation for Tracking Any Point

## Abstract

This paper tackles a challenge in learning the long-term point trajectories in videos, like the Tracking Any Point (TAP) task. Fundamentally, the estimation of point-level motions is hindered by the significant uncertainty inherent in comprehensive comparisons across the entire video frame. While existing models attempt to mitigate this issue by considering a regularized comparison space (*e.g.*, the cost volumes), point-level motion remains highly noisy, often leading to failures on individual points. To tackle the issue, our key idea is to jointly track multiple points within a given semantic object: since points in an object tend to move together on average, individual noise trajectories can be effectively marginalized, subsequently obtaining fine-grained motion information. Specifically, we predict the object mask using point-prompted segmentation provided by Segment Anything Models (SAM) and enhance the performance of existing models through a systematic two-stage procedure: (a) estimating the average motion of points within the object mask (predicted by SAM) as the initial estimate, and (b) refining this estimate to achieve point-level tracking. In stage (b), we actively generate fine-grained features around the initial estimate, preserving high-frequency details for precise tracking. Consequently, our method not only overcomes the failure modes seen in existing state-of-the-art methods but also demonstrates superior precision in tracking results. For example, on the recent TAP-Vid benchmark, our method advances the state-of-the-art performance, achieving up to a 25% improvement in accuracy at the 1-pixel error threshold. Furthermore, we showcase the advantages of our method in two downstream tasks: video depth estimation and video frame interpolation, exploiting the point-wise correspondence in each task.

## 1 Introduction

Finding point correspondences over the multiple views of a scene is a crucial challenge in handling visual data, playing a vital role in tasks understanding the geometry of the scene, *e.g.*, 3D reconstruction (Hartley & Zisserman, 2003). Specifically, when the scene features dynamically moving objects and backgrounds, *i.e.*, video data, the problem becomes estimating dense point-level motion over the video frames. For example, optical flow (Teed & Deng, 2020) has widely been adopted, where its goal is predicting the pixel-wise displacements between two frames, assuming that the majority of the pixels are visible in both frames. Consequently, the optical flow is only reliable within short-term frames, and its application can be limited to videos comprising longer frames.

To this end, video point tracking has recently emerged as a prominent next direction, overcoming the limitation of the previous approach. For example, the recent *Tracking Any Point* (TAP) task (Doersch et al., 2022) has attracted many research focus, which aims to predict the long-term trajectory of a given point, as well as the occlusion probability over the whole frames in a given video. However, tracking motions over the entire video entails comprehensive comparisons across the 3D spatial-temporal coordinates. In turn, the main challenge of point tracking is mostly regarding how to handle the computation burden and uncertainties arising from the large comparison space.

For instance, the canonical design in state-of-the-art models (Karaev et al., 2023; Doersch et al., 2023; Harley et al., 2022) considers the regularized space, referred to as the *cost volume* (Xu et al., 2017), which basically represents the likelihood of the point's location in low-resolution frames, producing a coarse-grained trajectory. Then, they employ refinement modules that smooth away

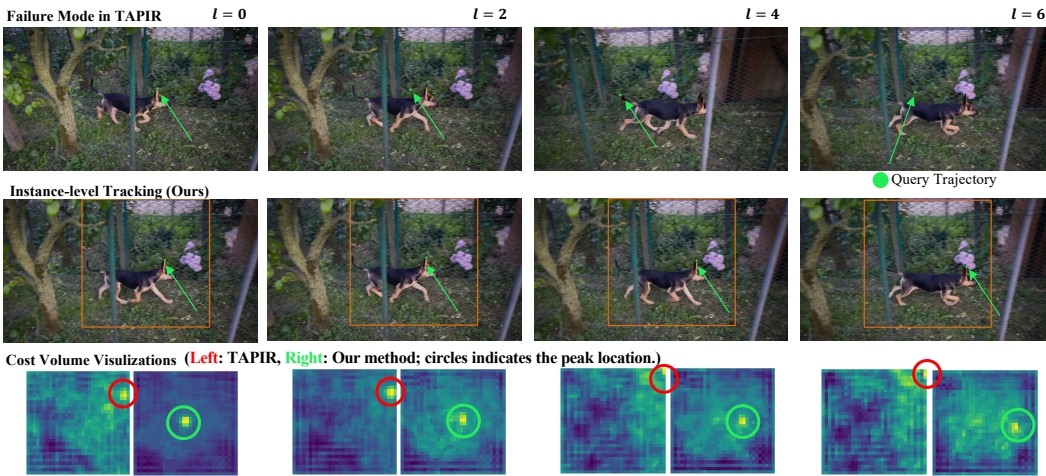

Figure 1: **The illustration of the failure mode in point tracking.** Even the state-of-the-art baseline, *e.g.*, Doersch et al. (2023), can fall into the failure mode when the cost volume (*i.e.*, likelihood of the point's location) is significantly erroneous; for example, when the model fails to capture the high-frequency visual details to represent thin object parts, such as the dog's ear. Our method mitigates this issue by referring to the estimated instance-level motion and focusing only on the region occupied by the instance to ensure the visual details required for accurate tracking. Indeed, we find that the cost volume in the baseline (left, red circles) is moving in the wrong direction to the query point, while our method (right, green circles) is correctly tracking the dog's ear.

the errors in the initial predictions, which are trained with datasets annotated for tracking, such as the Kubric (Greff et al., 2022) and PointOdyssey (Zheng et al., 2023). However, there exist failure modes in these methods, specifically when the cost volume is significantly erroneous and the refinement module fails to fix it, *e.g.*, the failure case on thin object parts in Fig. 1. We note that enhancing the quality of the cost volume is not trivial, as the naive extension of its size trades off its purpose of regularizing the comparison space and computational demands.

To overcome the fundamental limitation, our key idea is utilizing the semantic instance-level motion estimates to actively guide the tracking model to focus only on the important region when building the cost volume. Intuitively, the points on the same object instance are physically bonded, so that they would jointly share the long-term motion statistics. Hence, their aggregated motion can act as a reliable estimate. Specifically, we leverage the recent *Segment Anything Models* (SAM; Kirillov et al. 2023) to predict the segmentation mask of the instance that a given query point belongs to and utilize points on the segmentation mask when aggregating the instance-level motions.

Formally, we propose InstaTAP: **Insta**nce motion estimation for **T**racking **A**ny **P**oint, a new tracking method that overcomes the failure modes seen in existing state-of-the-art models, as-well-as providing a significant gain for models to be accurate up to the pixel scale. In a nutshell, InstaTAP can be built on top of any existing point tracker, *e.g.*, Doersch et al. (2023), to bootstrap its tracking performance. Specifically, given the segmentation mask indicating the semantic instance of the query point, we sample and track points on the mask and aggregate their trajectory. Then, we leverage this instance-level motion to actively clip the video frames along the instance's trajectory, so that the tracking model can only focus on the clipped regions and more details can be preserved in the cost volume without modifying the size.

Through the experiments on TAP-Vid point tracking benchmark suite (Doersch et al., 2022), we demonstrate the effectiveness of our new point tracking method using instance-level motions; for example, our method achieved up to 25% relative improvement in the 1-pixel tracking accuracy compared to the strongest baseline, Karaev et al. (2023). In addition, we present the application of our point tracking in two downstream tasks: robust video depth estimation (Kopf et al., 2021) and video frame interpolation (Chen & Jiang, 2023). Overall, our work highlights the effectiveness of considering instance-level motion as a reliable estimate for enhancing video point tracking, and we believe our work could inspire researchers to consider a new way to further leverage it in the future.

## 2 RELATED WORK

### 2.1 OPTICAL FLOW

The optical flow tackles the dense computation of instantaneous motion patterns between two given video frames. Commencing with the pioneering work of the FLowNet series which extended the adaptation of convolutional neural networks for motion estimation (Dosovitskiy et al., 2015; Ilg et al., 2017), DCFlow (Xu et al., 2017) introduced the innovative concept of the cost volume. This concept effectively encapsulates the dense correspondence between pairs of image patches. Subsequent breakthroughs, such as PWC-Net (Sun et al., 2018) and RAFT (Teed & Deng, 2020), have laid out systematic methodologies for processing the cost volume. These methodologies have since become integral to a plethora of modern motion estimation models, encompassing elements like feature pyramids, rigid warping, and iterative refinement techniques. Despite the progress in optical flow research, it is essential to acknowledge its inherent limitations, valid only within the context of a pair of frames, rendering it incapable of providing predictions regarding occlusions. This limitation assumes particular significance in the downstream video applications.

### 2.2 POINT TRACKING

The inherent constraint within optical flow has triggered the recent emergence of the point-tracking task. In essence, the task entails tracking any arbitrary physical point within a video sequence, encompassing both its occlusion behavior and the regression of its spatial coordinates across the entire duration of the video frames. Prominent contemporary models in this domain, such as PIPs (Harley et al., 2022), TAPNet (Doersch et al., 2022), TAPIR (Doersch et al., 2023), MFT (Neoral et al., 2023), and CoTracker (Karaev et al., 2023), all incorporate the foundational concept of the cost volume. Meanwhile, the distinguishing factor among these models is the refinement mechanisms employed subsequent to the initial estimation of correspondence within the cost volume. For instance, PIPs (Harley et al., 2022) utilize a recurrent neural network for iterative refinement, whereas both TAPNet (Doersch et al., 2022) and TAPIR (Doersch et al., 2023) leverage 3D convolutions and mixers to enhance the correspondence estimation. Nevertheless, the refinement modules can fail when the cost volume is extremely erroneous (see Fig. 1), and one of the focuses of this paper is preventing errors in the cost volume so that any existing tracking model can be improved.

### 2.3 PROMPTED POINT SEGMENTATION

A recent advancement within the domain of image segmentation has been the introduction of prompted segmentation tasks, notably spearheaded by the pioneering Segemnt Anything Model (SAM; Kirillov et al. (2023)). SAM is specifically designed to perform image segmentation on regions indicated by general point and box-prompted user queries. Bolstered by the extensive SA-1B dataset (Kirillov et al., 2023), which comprehensively represents the broad concept of segmentation within natural images and drawings. SAM and its derivative architectures (Zhang et al., 2023; Zhao et al., 2023; Ke et al., 2023), exhibit an impressive capacity for class-agnostic image segmentation. In the context of point tracking, specifically, prompted segmentation serves as a valuable resource by generating segmentation masks for physical surfaces for the object instance indicated by the query point. These masks enable our method to sample a set of coordinates that are subsequently utilized to estimate the instance-level motion.

## 3 METHOD

In this section, we provide the preliminary of tracking any point (TAP) task and our method details. Our idea leverages that the points on the same object instance are physically bonded, sharing the long-term motion statistics. Specifically, we directly leverage the foundational prior from recent point-queried segmentation models to segment object instances, indicating which segmentation mask a given query point belongs to. Then, we jointly track a set of points on the mask, referred to as the semantic neighbors, and take their expected motion to marginalize over individual noise trajectories. This instance-level motion can serve as an initial guess of the query trajectory, where we propose further steps for bootstrapping the tracking performance given the reliable initial guess, significantly boosting the tracking precision, which we refer to it as high-fidelity tracking.

The organization is as follows. In Secs. 3.1 and 3.2, we formally define the terminology and describe the canonical design found in point tracking literature, including the cost volume. Then, we provide the details of our method, step by step. First, we describe our instance-level motion estimation module in Sec. 3.3. For a given query point, this module predicts the initial guess of the query trajectory based on the joint instance-level motion estimate of the points within the point-queried segmentation mask. Next, in Sec. 3.4, we describe our enhanced high-fidelity point tracking module, which is designed to bootstrap the tracking performance given the initial guess by utilizing enhanced visual features gathered by constraining the tracking regions around the guessed trajectory.

## 3.1 BASIC FORMULATION

Point tracking is a task of predicting the point trajectory, given a query point in a sequence of video frames. When there exist multiple query points, *e.g.*, every pixel within a frame, they can be jointly tracked through batch-processing in practice, so we focus on a single-point formulation without loss of generality.

Formally, let $\mathsf{I} \in \mathbb{R}^{H \times W \times L \times 3}$ be an RGB, monocular video frame sequence of the resolution $(H, W)$ and the frame length $L$, and $\boldsymbol{p} \equiv (x, y, t) \in (0, 1)^3$ be the continuous point coordinate in the (horizontal, vertical, temporal) basis within the video. To give an example for clarity, $(0, 0, 0)$ denotes the left-top corner in first frame, $(1, 1, 1)$ denotes the right-bottom in final frame, and the center of the discrete pixel $\mathsf{I}_{h,w,l}$ is equivalent to $((w + 0.5)/W, (h + 0.5)/H, (l + 0.5)/L)$. Also, let $\boldsymbol{T} \in (0, 1)^{L \times 2}$ represent the predicted trajectory over the entire video frames, and $\boldsymbol{o} \in (0, 1)^L$ represent the probability of occlusion for each frame, associated with the trajectory.

We consider a model $\mathtt{Tracker}$, which takes the video frame $\mathsf{I}$ and the query point as the inputs, denoted by $\boldsymbol{p}^{(\mathsf{q})} \in (0, 1)^3$, and predicts the trajectory $\boldsymbol{T}^{(\mathsf{q})} \in (0, 1)^{L \times 2}$ and the occlusion probability $\boldsymbol{o}^{(\mathsf{q})} \in (0, 1)^L$ over the entire set of frames,

$$(\boldsymbol{T}^{(\mathsf{q})}, \boldsymbol{o}^{(\mathsf{q})}) := \mathtt{Tracker}(\boldsymbol{p}^{(\mathsf{q})}, \mathsf{I}). \tag{1}$$

## 3.2 COST VOLUME ARCHITECTURE

In the canonical design of contemporary tracking models, the key component is the cost volume, which basically represents the likelihood of the spatial-temporal location of the query point in the video, denoted by $\mathbf{C} \in (-1, 1)^{H_F \times W_F \times L}$, where $(H_F, W_F)$ is the resolution of the cost map. For example, the element $\mathbf{C}_{h,w,l}$ represents the likelihood that the query point is on that specific discrete location. The specific process of building the cost volume is as follows.

First, the video $\mathsf{I}$ is processed into the regular-sized visual feature map $\mathbf{F} \in \mathbb{R}^{H_F \times W_F \times L \times D}$, *e.g.*, by convolutional networks with pooling layers. Here, each feature vector $\mathbf{F}_{h,w,l}$ is considered the embedding of its corresponding point coordinate. In general, to support the continuous coordinate $\boldsymbol{p} \in (0, 1)^3$, the point embedding is gathered by interpolating the adjacent feature vectors (*e.g.*, bilinear interpolation), which we denote $\hat{F}(\boldsymbol{p}) \in \mathbb{R}^D$.

Next, for the given query point $\boldsymbol{p}^{(\mathsf{q})}$ and its embedding $\hat{F}(\boldsymbol{p}^{(\mathsf{q})})$, the similarity with respect to the feature map $\mathbf{F}$ is calculated, constituting each element of the cost volume, $\mathbf{C}_{x,y,t} := S\left(\mathbf{F}_{x,y,t}, \hat{F}(\boldsymbol{p}^{(\mathsf{q})})\right)$, where $S(\cdot, \cdot)$ is the vector similarity (*e.g.*, dot product). Finally, models predict the coarse trajectory estimates by connecting the elements with the largest similarity for each frame, and also detect occlusions by thresholding the similarity values. Subsequently, they produce the final tracking results by improving the smoothness of the coarse trajectory through sophisticated refinement mechanisms.[1]

We note that one of our main focuses is to prevent failure modes in the cost volume, as exemplified in the Fig. 1. For example, the cost volume can be significantly erroneous when the embedding $\hat{F}(\boldsymbol{p}^{(\mathsf{q})})$ lacks the high-frequency details required to correctly represent the visual details in the query point, which causes a common tracking failure mode.

---

[1] We refer the readers to literature for the refinement mechanisms found in practical models, *e.g.*, Karaev et al. (2023); Doersch et al. (2023)

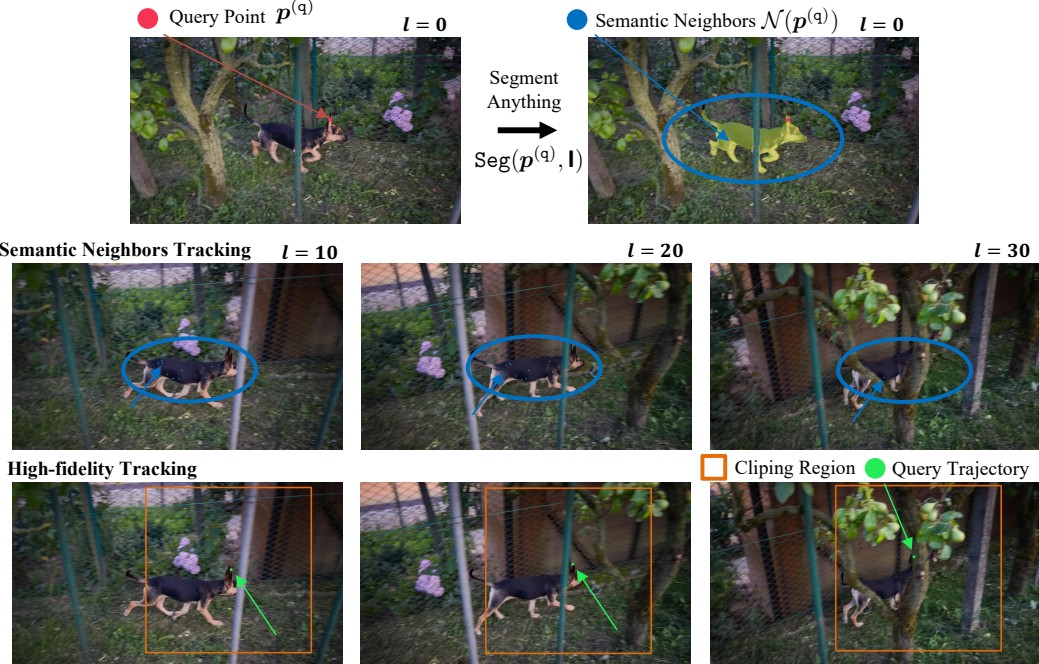

Figure 2: **Illustration of our point tracking mechanism.** In the initial frame ($l = 0$) and the given query point $\boldsymbol{p}^{(\mathsf{q})}$ (the red dot), we predict the semantic object mask (depicted with the yellow mask) using the Segment Anything Model (SAM) and sample the semantic neighbors (the blue dots) from the mask. By tracking the semantic neighbor, we estimate the instance-level motion, which essentially locates the clipping region (the orange square). Finally, the query trajectory (the green dot) is predicted within the clipping region, *i.e.*, the high-fidelity tracking.

### 3.3 INSTANCE-LEVEL MOTION ESTIMATION

Given a query point suffering from the failure modes, our key idea is to compute the instance-level motion estimate and then incorporate it into the point-level tracking process. Intuitively, instance-level tracking could be viewed as a relaxed problem compared to point-level tracking as marginalizing individual point-level sources would reduce noise. Additionally, instance-level motion is beneficial for tracking individual points, as points within the same object instance are physically connected and tend to exhibit similar long-term motion statistics.

Specifically, our method tracks a set of points contained in the instance mask predicted by a point-query segmentation model, *e.g.*, Ke et al. (2023), which produces the pixel-wise confidence in how the pixels are semantically related to the query point $\boldsymbol{p}^{(\mathsf{q})}$. For example, we consider the model Seg, which produces the segmentation mask for the initial frame indicated by the query's time coordinate,

$$\texttt{Seg}(\boldsymbol{p}^{(\mathsf{q})}, \mathsf{l}) := (0, 1)^{H \times W}. \tag{2}$$

Specifically, we sample a fixed $N$ number of points of based on $\texttt{Seg}(\boldsymbol{p}^{(\mathsf{q})}, \mathsf{l})$. For the confidences having significantly large values, *e.g.*, $\texttt{Seg}(\boldsymbol{p}^{(\mathsf{q})}) \geq 0.5$, we employ a weighted multinomial sampling to choose a set of points, coined semantic neighbors, $\mathcal{N}(\boldsymbol{p}^{(\mathsf{q})}) := \{\boldsymbol{p}^{(a_0)}, \dots, \boldsymbol{p}^{(a_N)}\}$, where $\boldsymbol{p}^{(a_0)} \equiv \boldsymbol{p}^{(\mathsf{q})}$ for convenience.

Then, we jointly track these semantic neighbors using $\texttt{Tracker}$ and the frame-wise trajectory displacements,

$$\left(\boldsymbol{T}^{(a_i)}, \boldsymbol{o}^{(a_i)}\right) = \texttt{Tracker}\left(\boldsymbol{p}^{(a_i)}, \mathsf{l}\right), \tag{3}$$

$$\Delta\boldsymbol{T}_{t,:}^{(a_i)} := \boldsymbol{T}_{t,:}^{(a_i)} - \boldsymbol{T}_{t-1,:}^{(a_i)} \text{ for } t \geq 1. \tag{4}$$

Here, some semantic neighbors could fall into failure modes, while others tend to be accurately predicted showing minimal occlusion probabilities $o^{(a_i)}$. Hence, we perform a weighted aggregation with their occlusion confidences, so the effect of invisible trajectories can be removed, producing a reliable instance-level motion estimates,

$$\Delta \bar{\boldsymbol{T}}_{t,:}^{(\mathsf{q})} := \sum_{i=0}^{N} \frac{(1 - o_t^{(a_i)}) \cdot \Delta \boldsymbol{T}_{t,:}^{(a_i)}}{\sum_{j=0}^{N}(1 - o_t^{(a_j)})}, \tag{5}$$

$$\bar{\boldsymbol{T}}_{t,:}^{(\mathsf{q})} = \bar{\boldsymbol{T}}_{t-1,:}^{(\mathsf{q})} + \Delta \bar{\boldsymbol{T}}_{t,:}^{(\mathsf{q})} \ \text{ for } t \geq 1, \tag{6}$$

where $\bar{\boldsymbol{T}}_{0,:}^{(\mathsf{q})} \equiv \boldsymbol{p}^{(\mathsf{q})}$. While the above aggregation formula mostly functions as expected, we could identify a seldom edge case due to the failure of the segmentation model; the mask might not indicate the correct instance-level region. Nevertheless, we find the problem can be mitigated by simply putting a stronger preference to the query trajectory, in an adaptive manner depending on its occlusion probability,

$$\hat{\boldsymbol{T}}_{t,:}^{(\mathsf{q})} := (1 - o_t^{(\mathsf{q})}) \cdot \boldsymbol{T}_{t,:}^{(\mathsf{q})} + o_t^{(\mathsf{q})} \cdot \bar{\boldsymbol{T}}_{t,:}^{(\mathsf{q})}. \tag{7}$$

Finally, our model can produce the reliable initial guess for the instance-level trajectory even when affected by the failure mode of the cost volume, given a sufficiently large constant $N$, *e.g.*, we choose $N = 31$ unless stated otherwise.

### 3.4 ENHANCED HIGH-FIDELITY POINT TRACKING

Given the instance-level motion estimate, we now aim to utilize it for enhanced point-tracking performance. Specifically, we propose to actively generate fine-grained features around the estimated trajectory, preserving high-frequency details for precise tracking.

Recall that the feature map $\mathbf{F}$ has the fixed resolution $(H_F, W_F)$, as described in Sec. 3.2. Specifically, the existing models perform the image resizing to shrink the resolutions to meet the resolution requirement, which essentially acts as the low-pass filters that sweep away fine-grained details from the feature maps. As there is a good initial trajectory guess in our method, we propose to actively clip each frame, around the region centered at the trajectory.

Specifically, let $(H_C \times W_C)$ a clipping resolution with conditions $H_C < H$ and $W_C < W$, and we denote the active clip for the query trajectory as

$$\mathbf{I}^{(\mathsf{q})} := \texttt{clip}(\mathbf{I}, \hat{\boldsymbol{T}}^{(\mathsf{q})}, H_C, W_C) \in \mathbb{R}^{H_C \times W_C \times L \times 3}. \tag{8}$$

Given the clipped frames, the degree of resizing in the feature map reduces by $(H_C \times W_C)/(H \times W)$, and thus more high-frequency details can be preserved in the visual features. We would like to emphasize that the feature resolution $(H_F \times W_F)$ is not modified by clipping, hence the size of the cost volume remains the same. Specifically, we denote this enhanced version of the tracker by

$$(\tilde{\boldsymbol{T}}^{(q)}, \tilde{\boldsymbol{o}}^{(\mathsf{q})}) := \texttt{TrackerHD}(\boldsymbol{p}^{(\mathsf{q})}, \mathbf{I}^{(\mathsf{q})}). \tag{9}$$

Additionally, we can boost the performance of `TrackerHD`, by considering the progressive reduction of the clip sizes $(H_k, W_k)$. Specifically, we define the recursive predictions,

$$(\tilde{\mathbf{T}}_{k+1}^{(\mathsf{q})}, \tilde{\boldsymbol{O}}_{k+1}^{(\mathsf{q})}) := \texttt{TrackerHD}_k(\boldsymbol{p}^{(\mathsf{q})}, \texttt{clip}(\mathbf{I}, \tilde{\mathbf{T}}_k^{(\mathsf{q})}, H_k, W_k)) \ \text{ for } k \geq 0, \tag{10}$$

where $\mathbf{T}_0^{(\mathsf{q})} \equiv \hat{\boldsymbol{T}}^{(\mathsf{q})}$. In this recursive structure with the progressive resolution reduction, the model is essentially being bootstrapped to achieve a better tracking performance. Finally, we note that it is also useful to aggregate the outputs from the progressive clip sizes, giving an effect similar to the multi-scale pyramids found in various vision architectures (Lin et al., 2017; Liu et al., 2016). To do so, we perform the weighted aggregation of the trajectories by the occlusion probabilities, and the harmonic mean for the occlusion probabilities to get the final prediction:

$$\boldsymbol{T}_{t,:}^{(\mathsf{q})} := \sum_{k=0}^{K} \frac{1 - \tilde{O}_{k,t}^{(\mathsf{q})} \cdot \tilde{\mathbf{T}}_{k,t,:}^{(\mathsf{q})}}{\sum_{j=0}^{K} \tilde{O}_{j,t}^{(\mathsf{q})}} \in (0,1)^2, \tag{11}$$

$$o_t^{(\mathsf{q})} := \left( \prod_{k=0}^{K} \tilde{O}_{k,t}^{(\mathsf{q})} \right)^{\frac{1}{K+1}} \in (0,1). \tag{12}$$

Table 1: **TAP-Vid benchmark results on high-precision metrics.** The $\delta_n^x$ indicates the ratio of the correct trajectories (*i.e.*, accuracy), judged by whether it is within $n$ pixels from the ground truth, and the $J_n$ (Jaccard-n) indicates the ratio of the correct trajectories with the true positive visibility prediction.

| Method | DAVIS-F | | | | DAVIS-S | | | |
|---|---|---|---|---|---|---|---|---|
| | $J_1$ | $\delta_1^x$ | $J_2$ | $\delta_2^x$ | $J_1$ | $\delta_1^x$ | $J_2$ | $\delta_2^x$ |
| TAPNet (2022) | 20.7 | 30.1 | 40.4 | 51.7 | 25.3 | 36.3 | 46.0 | 57.8 |
| PIPS2 (2023) | 19.6 | 35.8 | 35.1 | 57.8 | 6.9 | 14.2 | 14.2 | 27.0 |
| TAPIR (2023) | 23.0 | 34.3 | 45.0 | 58.5 | 28.1 | 41.0 | 51.4 | 65.3 |
| CoTracker (2023) | 28.3 | 43.5 | 49.7 | 67.0 | 34.9 | 50.9 | 55.4 | 71.9 |
| **InstaTAP (Ours)** | **35.4** | **48.2** | **57.7** | **70.1** | **41.9** | **55.0** | **63.6** | **75.2** |
| | Kinetics | | | | RGBStack | | | |
| TAPNet (2022) | 18.9 | 28.3 | 37.3 | 48.8 | 38.9 | 55.6 | 58.8 | 74.1 |
| PIPS2 (2023) | 14.0 | 28.6 | 28.0 | 50.5 | 29.8 | 50.6 | 45.0 | 68.6 |
| TAPIR (2023) | 17.9 | 27.4 | 36.7 | 49.0 | 38.5 | 55.7 | 59.6 | 75.6 |
| CoTracker (2023) | 20.3 | 33.3 | 36.8 | 53.4 | 35.4 | 52.3 | 52.3 | 70.9 |
| **InstaTAP (Ours)** | **23.8** | **34.0** | **42.6** | **54.6** | **44.2** | **60.2** | **61.5** | **76.5** |

## 4 EXPERIMENTS

In this section, we demonstrate the effectiveness of the proposed method, InstaTAP. Specifically, we mainly incorporate our method with the recent point tracking and refinement model, TAPIR (Doersch et al., 2023).[2] For the segmentation model, we utilize MobileSAM (Ke et al., 2023) a lightweight model variant trained in the same dataset as the original SAM (Kirillov et al., 2023). We choose the `TrackerHD` hyperparameters $H_0 = W_0 = 960$, $H_1 = W_1 = 384$ and $K = 1$, for all experiments unless stated otherwise. We note that the frame resolutions are not affected by these hyperparameters, as models always resize images to a pre-defined size; the clipping only controls the size of physical regions depicted by the frames (Creamer, 2010).

### 4.1 TRACKING ANY POINT

**Baselines.** We compare our method to the recent baselines CoTracker (Karaev et al., 2023), TAPIR (Doersch et al., 2023), PIPS2 (Zheng et al., 2023), and TAPNet (Doersch et al., 2022), by utilizing the official checkpoints and hyperparameters provided in their project pages.

**Datasets.** We evaluate these models in three different datasets from the TAP-Vid benchmark suit (Doersch et al., 2022), DAVIS (Pont-Tuset et al., 2017), Kinetics (Carreira & Zisserman, 2017), and RGB-Stacking (Lee et al., 2021), each of which represents distinct characteristics. For example, DAVIS (Pont-Tuset et al., 2017) contains 30 videos, specifically curated for evaluating the tracking performance under the large variance in the appearance and motions of the object entities. Its two variants, DAVIS-F and DAVIS-S differ by how the query points are given to models: DAVIS-F queries the model only once in the first frame, while DAVIS-S queries the model in strides of five frames. As DAVIS-F requires long-term tracking, it is generally a more difficult setting. Kinetics (Carreira & Zisserman, 2017) contains 1,144 web videos collected from YouTube representing realistic noisy characteristics of the video in the wild, such as sudden scene changes. RGB Stacking (Lee et al., 2021) is a synthetically rendered dataset, depicting 50 different moves by a robotic arm.

**Point tracking.** To measure the quality of point tracking, the benchmark considers the $\delta_n^x$ accuracy which indicates the ratio of the correct trajectories judged by whether it is within n-pixel error threshold around the ground truth. In addition, $J_n$ (Jaccard-n) metric takes the model's occlusion prediction into account, by measuring the ratio of the correct points with the true positive visibility predictions.

---

[2]Our method can be generally applied to any existing models, even including the concurrent CoTracker model (Karaev et al., 2023) that is available very recently at the moment this work is almost done. Nevertheless, ours with TAPIR outperforms CoTracker and we did not put efforts to incorporate our method with CoTracker at this submission (although ours + CoTracker is expected to perform even better than ours + TAPIR).

Table 2: **Average Metrics in Tap-VID Benchmark.** The $\delta_{avg}^x$ indicates the average ratio of the trajectory being within different error thresholds (1, 2, 4, 8 and 16 pixels), and AJ (Average Jaccard) additionally considers whether the visibility of the trajectory is true positive.

| Method | DAVIS-F AJ | DAVIS-F $\delta_{avg}^x$ | DAVIS-S AJ | DAVIS-S $\delta_{avg}^x$ | Kinetics AJ | Kinetics $\delta_{avg}^x$ | RGBStack AJ | RGBStack $\delta_{avg}^x$ |
|---|---|---|---|---|---|---|---|---|
| TAPNet (Doersch et al., 2022) | 51.6 | 63.8 | 56.5 | 68.2 | 49.2 | 60.6 | 65.4 | 79.0 |
| PIPS2 (Zheng et al., 2023) | 46.6 | 69.4 | 25.6 | 42.9 | 37.3 | 62.0 | 52.3 | 74.9 |
| TAPIR (Doersch et al., 2023) | 57.5 | 70.5 | 62.8 | 75.1 | 50.2 | 62.3 | 66.3 | 80.6 |
| CoTracker (Karaev et al., 2023) | 60.8 | 76.1 | 64.3 | 78.9 | 48.2 | 64.4 | 64.1 | 78.0 |
| **InstaTAP (Ours)** | **65.3** | **78.6** | **69.3** | **81.4** | **51.4** | **65.8** | **66.6** | **81.8** |

Table 3: **Ablation study of the components in our model.** We ablate the effect of Instance-level Motion (IM), active feature clipping (Clip), and the multi-scale fusion (Fusion) modules considered in our method. For the evaluation, we calculate both pixel-scale and average-scale metrics under the DAVIS-F dataset (Doersch et al., 2022).

| IM | Clip | Fusion | $J_1$ | $\delta_1^x$ | AJ | $\delta_{avg}^x$ |
|---|---|---|---|---|---|---|
| ✗ | ✗ | ✗ | 23.0 | 34.3 | 57.5 | 70.5 |
| ✓ | ✗ | ✗ | 28.2 | 41.1 | 62.5 | 75.3 |
| ✓ | ✗ | ✓ | 28.3 | 41.2 | 62.6 | 75.6 |
| ✓ | ✓ | ✗ | 34.0 | 48.0 | 62.9 | 77.0 |
| ✓ | ✓ | ✓ | **35.4** | **48.2** | **65.3** | **78.6** |

First of all, we demonstrate the efficacy of our InstaTAP in the high-precision (*i.e.*, 1- and 2-pixel error thresholds) metrics in Tab. 1. For example, InstaTAP can achieve up to 25% relative improvements in the Jaccard-1 metric $J_1$ in DAVIS-F, compared to the strongest baseline CoTracker(Karaev et al., 2023). We attribute this gain to the enhanced high-frequency details by our active feature enhancement module. To provide a more general comparison with respect to the baselines, we also provide the evaluation results of the models in the average metrics in Tab. 2, which represents the overall tracking performance in different precision thresholds. We find our method can still outperform every baseline, *e.g.*, 65.3% (ours) vs. 60.8% (Karaev et al., 2023). Overall, we argue that our method demonstrates a reliable performance, generalizing well to various datasets (Pont-Tuset et al., 2017; Carreira & Zisserman, 2017; Lee et al., 2021) and precision thresholds.

**Ablation study.** We additionally perform an ablation study to understand further how each component in our method affects the performance in Tab. 3. Specifically, we consider the instance-level motion estimation (IM), the active feature enhancement by clipping (Clip), and the multi-scale fusion (Fusion) as the subject for the ablation. Following our key motivation, incorporating the instance-level motion provides the most significant effect, (*e.g.*, 23.0 → 28.2 in Jaccard-1), and the feature clipping provides another comparable gain, (*e.g.*, 28.2 → 35.4 in Jaccard-1). The multi-scale fusion provides some additional boost, (*e.g.*, 34.0 → 35.4 in Jaccard-1) when combined with the feature clipping. We note that the multiscale fusion without the feature clipping has only a minor ensemble effect on the model's output, and thus its effect becomes marginal.

## 4.2 DOWNSTREAM APPLICATIONS

The point-level motion estimates in monocular videos can be applied to various downstream applications suffering from the lack of accurate correspondence. For example, the consistent video depth estimation (Zhang et al., 2021; Luo et al., 2020; Kopf et al., 2021) and the video frame interpolation (Chen & Jiang, 2023; Jin et al., 2023).

**Consistent video depth estimation**. In a monocular (single-frame) depth estimation, predictions are essentially constrained to be only accurate up-to-scale (Hartley & Zisserman, 2003). As a result, they suffer from the large variance in the prediction scales over the frames, when directly applied to videos. To mitigate the problem, the test-time optimization (Zhang et al., 2021; Kopf et al., 2021) calibrates the depth scales over the frames given externally provided pixel-wise correspondences. Specifically, we test our method on R-CVD optimizer (Kopf et al., 2021), which minimizes the

Table 4: **The effect of point tracking in consistent depth estimation.** We execute the R-CVD depth optimizer (Kopf et al., 2021) on the motion estimates by the optical flow (Jin et al., 2023) and our method. We utilize the test metrics of R-CVD, *e.g.*, $\delta^1$ accuracy with the threshold of 1.25m errors, and $L^1$ and RMSE representing the relative and absolute regression errors, respectively.

| Model | $\delta^1$ accuracy (%) ↑ | $L^1$ relative error (%) ↓ | RMSE (meters) ↓ |
|---|---|---|---|
| Flow (Jin et al., 2023) | 38.69 | 102.8 | 7.85 |
| InstaTAP (Ours) | 39.14 (+0.45) | 98.79 (-4.09) | 7.82 (-0.03) |

Table 5: **The effect of point tracking in video frame interpolation.** We evaluate the frame interpolation on SportsEBME (Chen & Jiang, 2023), an end-to-end trained baseline with reconstruction error, and the variants using the motion estimates by an optical flow (Jin et al., 2023) and our method, respectively. To measure the quality, we consider the standard metrics, *e.g.*, PSNR, structural similarity (SSIM; Wang et al. 2004), and RGB interpolation errors (IR-RGB; Baker et al. 2011).

| Model | PSNR (dB)↑ | SSIM (%) ↑ | IE-RGB $\in [0, 255]$ ↓ |
|---|---|---|---|
| SportsEBME (Chen & Jiang, 2023) | 26.48 | 89.73 | 15.67 |
| SportsEBME + Flow (Jin et al., 2023) | 26.05 (-0.33) | 88.94 (-0.79) | 16.70 (+1.03) |
| SportsEBME + InstaTAP (Ours) | 26.78 (+0.30) | 90.18 (+0.45) | 14.39 (-1.28) |

depth variance among the pixels associated with optical flow (Teed & Deng, 2020). As the optical flow is limited to two adjacent frames, we expect that our method can improve the consistency in the long term. Specifically, in Tab. 4, our method can achieve better depth consistency compared to state-of-the-art optical flow (Jin et al., 2023), *e.g.*, 4.09 points reduction in the $L^1$ relative error.

**Video frame interpolation**. Synthesizing novel intermediate frames from input images is a widely adopted subject of research in video processing, to enhance the clarity of the video content. Specifically, its recent paradigm introduces warping functions, where the images are synthesized through the learned perspective transform between adjacent frames, *e.g.*, softmax splatting (Niklaus & Liu, 2020). However, when the given frames feature complex motion and occlusion, the synthesized image can suffer from aliasing effects, as the model cannot correctly capture the pixel-wise correspondences. For example, the recent SportsSlomo benchmark (Chen & Jiang, 2023) is towards the application in dynamic sports videos, in which the model is largely affected by complex motions. In this regard, we check the efficacy of our point tracking method, by additionally providing pixel-wise motions to the splatting function found in a contemporary model in literature (Chen & Jiang, 2023). Specifically, in Tab. 5, we evaluate the frame interpolation quality of the vanilla model, a model provided with the external optical flow (Jin et al., 2023), and the model provided with point tracking by our method. As a result, we find that our method can improve the frame interpolation quality; for example +0.30dB gain in PSNR. Interestingly, we find the optical flow degrades the quality, *e.g.*, -0.33dB in PSNR, due to severe noises in its estimates under complex dynamic sports videos.

## 5 CONCLUSION

In this paper, we propose a new video point tracking method, coined InstaTAP, which aims to overcome the failure modes arising from the lack of high-fidelity details in considering the cost volume architecture, by incorporating instance-level motion estimates into point tracking. Specifically, we utilize the segmentation mask predicted by Segment Anything Models (SAM) as the object instance prior and jointly track the sampled points on the mask to estimate the instance-level motion. Subsequently, we utilize the instance-level motion to actively clip the video frames along the instance's trajectory, so that the tracking model can only focus on the clipped regions and more details can be preserved in the cost volume without extending the size. InstaTAP demonstrates a significant impact over the state-of-the-art and shows that our point tracking can be applied in downstream tasks such as video depth estimation and video frame interpolation, replacing the optical flow traditionally used in these models. Overall, our work highlights the effectiveness of considering instance-level motion as a reliable estimate for enhancing video point tracking, and we believe our work could inspire researchers to consider a new alternative way to further leverage it in the future.

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
