# A   HYPERPARAMETER ABLATION

Table 6: **Ablation study of the clipping size in our model.** We ablate the clipping size considered in our method. For the evaluation, we calculate both pixel-scale and average-scale metrics under the DAVIS-F dataset (Doersch et al., 2022).

| Size | $J_1$ | $\delta_1^x$ | AJ | $\delta_{\text{avg}}^x$ |
|------|-------|--------------|------|-------------------------|
| 960 | 29.8 | 42.4 | 63.1 | 75.8 |
| 768 | 31.9 | 44.6 | 64.0 | 76.6 |
| 384 | **35.7** | **47.6** | 62.7 | **76.3** |

Table 7: **Ablation study of the effect of number of semantic neighbors in our method.** We ablate the clipping size considered in our method. For the evaluation, we calculate both pixel-scale and average-scale metrics under the DAVIS-F dataset (Doersch et al., 2022).

| N | $J_1$ | $\delta_1^x$ | AJ | $\delta_{\text{avg}}^x$ |
|-----|-------|--------------|------|-------------------------|
| 127 | 35.1 | 44.9 | 62.3 | 75.7 |
| 63 | 35.1 | 45.0 | 62.4 | 75.7 |
| 31 | **35.7** | **47.6** | 62.7 | 76.3 |
| 15 | 34.3 | 47.3 | **64.3** | **76.7** |
| 7 | 31.8 | 44.6 | 64.2 | 76.6 |
| 3 | 28.2 | 41.2 | 62.1 | 75.1 |

In this section, we ablate the choice of hyperparameters in our method, namely the number of semantic neighbors $N$, considered in Eq. (2), and the clipping sizes $(H_C, W_C)$ in Eq. (8). We note that we only consider a squares upon clipping, using the identical $H_C$ and $W_C$ for all settings.

In Tab. 6, we find that using a smaller size as the clipping size introduces positive effects in the pixel-scale (*i.e.*, 1-pixel) performances, while it also trades off the average-scale performances. This is expected behavior, as the clipping size gets smaller, more high-frequency details of the visual feature would be preserved, but at the same time, the chance of erroneously focusing on a wrong region increases due to a limited clipping size. Through the multi-scale fusion considered in this paper, we effectively handle this trade-off and achieve optimal performance.

Next, in Tab. 7, we ablate the effect of the choice for the number of semantic neighbors $N$ (excluding the query point itself), halving down its value starting from $N + 1 = 128$ to $N + 1 = 4$. As a result, we find that the range $N \geq 15$ can provide satisfactory performance in our method, although there exists a mild trade-off between the pixel scale and the average scale performances depending on the choice. As one of our focuses is achieving optimal pixel scale performances in point tracking, we empirically choose $N = 31$ in this paper.

# B   QUALITATIVE RESULTS

In this section, we additionally provide the qualitative results of our tracking method on the challenging "libby" sample of the DAVIS-F dataset in Fig. 3 and Fig. 4.

The libby sample requires to track fine-grained object parts of a dog, such as its ears and tail. Examining qualitatively, we find our method is effectively tracking the hard examples. For example, in Fig. 3, our method can precisely detect the occlusion of the very tiny and bright side of the dog's ear (*i.e.*, the original query point), thanks to the high-frequency details preserved by our progressive clipping. Also, in Fig. 4, our method can also precisely track the very tiny tail of the dog (*i.e.*, the original query point).

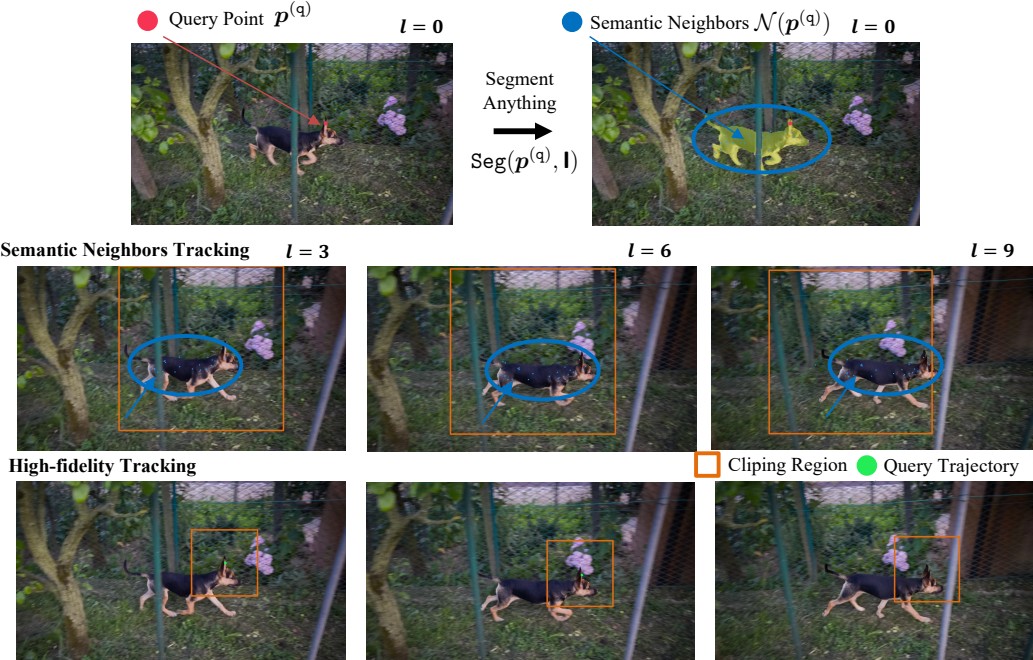

Figure 3: **Qualitative results in a hard "ear" tracking of libby in DAVIS-F.** In the initial frame ($l = 0$) and the given query point $p^{(q)}$ (the red dot), we predict the semantic object mask (depicted with the yellow mask) using the Segment Anything Model (SAM) and sample the semantic neighbors (the blue dots) from the mask. By tracking the semantic neighbor, we estimate the instance-level motion, which essentially locates the clipping region (the orange square). Finally, the query trajectory (the green dot) is predicted within the clipping region, *i.e.*, the high-fidelity tracking.

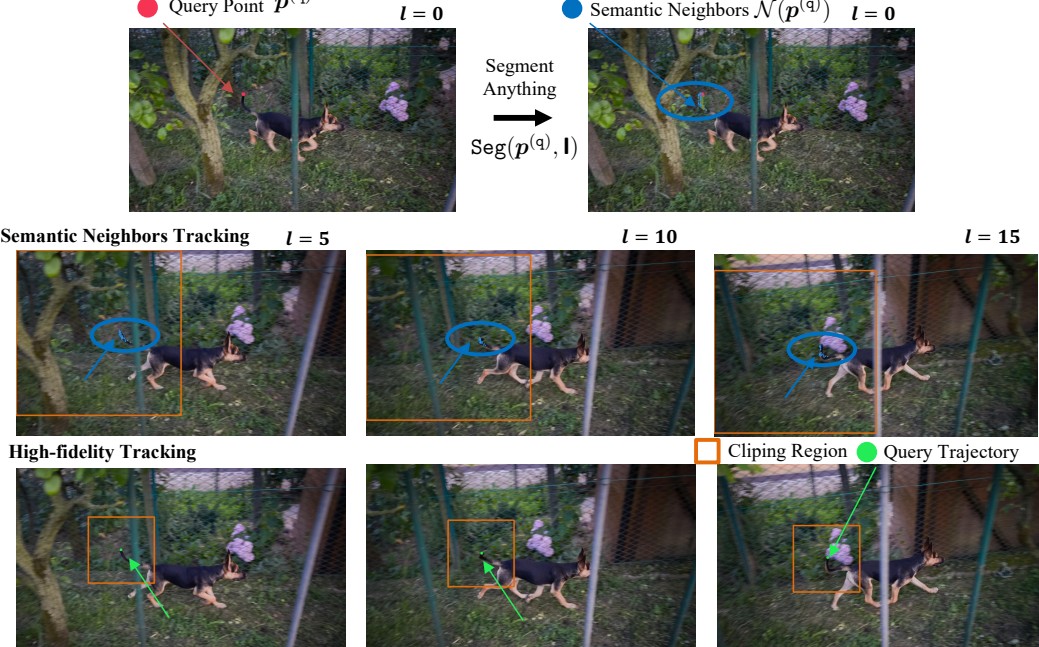

Figure 4: **Qualitative results in a hard "tail" tracking of libby in DAVIS-F.** In the initial frame ($l = 0$) and the given query point $p^{(q)}$ (the red dot), we predict the semantic object mask (depicted with the yellow mask) using the Segment Anything Model (SAM) and sample the semantic neighbors (the blue dots) from the mask. By tracking the semantic neighbor, we estimate the instance-level motion, which essentially locates the clipping region (the orange square). Finally, the query trajectory (the green dot) is predicted within the clipping region, *i.e.*, the high-fidelity tracking.