# OpenReview forum: "InstaTAP: Instance Motion Estimation for Tracking Any Point"
_ICLR.cc/2024/Conference — ICLR 2024 Conference Withdrawn Submission_

### Official Review · Reviewer_sKxk · 2023-10-30

**Soundness:** 3 good
**Presentation:** 3 good
**Contribution:** 2 fair
**Rating:** 3
**Confidence:** 4

**Summary:**

This paper is about tracking every point (TAP). Basically, it adopts a point-prompted segmentation by Segment Anything Model(SAM) to enhance the performance of existing models by estimating the average motion within the segmentation mask followed by a refinement stage to get final tracking results. Finally the authors evaluated on TAP-Vid benchmark to compare with previous published methods and show its practical usage in other vision tasks.

**Strengths:**

1. The paper is in general well-written. It clearly pinpoints the limitations of existing methods, e.g. the failure cases of cost volume calculation, then proceed to the proposed method part.

2. The idea of utilizing occlusion information within weighted aggregation, is reasonable, and demonstrates to be effective in the experiments.

**Weaknesses:**

1. I am not a fun of the overall idea. Basically, this work is about an "A+B" style and I don't see too much innovation behind simply adding segmentation masks produced by SAM model. If SAM can be used, then any other SOTA semantic/instance segmentation models can be adopted to yield better performance. So the contribution here is a bit too trivial.

2. There are too many engineering stuffs in designing the tracker. For example, clipping, multi-scale operations. I don't see any "learning" stuff in the contribution side.

3. Overall misleading claim. For example, in the abstract part, authors claim a SOTA performance " For example, on the recent TAP-Vid benchmark, our method advances the state-of-the-art performance," But the performance is at least inferior to the paper [A] in TAP-Vid. For example, In Kinetics [A] has AJ 55.1, $\sigma_{avg}$69.6 while this work achieves only AJ 51.4, $\sigma_{avg}$65.8. On RGB-Stacking [A] has AJ77.5 and $\sigma_{avg}$87.0 while this work gets AJ66.6 and $\sigma_{avg}$81.8, this is also the case for DAVIS dataset. That is to say, the performance is NOT SOTA indeed, and it is hard to justify the usefulness of the overall idea.

References:
[A] Qianqian Wang et.al. Tracking Everything Everywhere All at Once. ICCV2023

**Questions:**

I wonder is it possible to add other metrics in TAP-Vid, such as OA, TC, for a more complete comparison with prior methods?

---

### Official Review · Reviewer_K1PM · 2023-11-01

**Soundness:** 3 good
**Presentation:** 3 good
**Contribution:** 2 fair
**Rating:** 5
**Confidence:** 4

**Summary:**

This study tackles the challenge of accurately learning long-term trajectories for individual points in video sequences, such as the Tracking Any Point (TAP) task. Point-level motion estimation is hindered by the inherent uncertainty in comprehensive frame-wide comparisons. Existing models address this by using regularized comparison spaces like cost volumes, but they still suffer from noisy point-level motion, leading to tracking failures. To overcome this, the proposed method jointly tracks multiple points within a semantic object, leveraging the fact that points within an object tend to move together. By predicting object masks with Segment Anything Models (SAM) and implementing a two-stage procedure, the approach significantly improves tracking precision, surpassing state-of-the-art methods by up to 25% in accuracy on the TAP-Vid benchmark. Additionally, the approach demonstrates advantages in video depth estimation and frame interpolation by utilizing point-wise correspondence in these tasks.

**Strengths:**

Using Segment Anything Models (SAM) to enhance performance is a very interesting idea for motion prediction.

**Weaknesses:**

1. The main contribution is to exploit SAM to boost the performance of tracking. However, the SAM is an existing method so the contribution of this framework is limited.

2. The proposed method's performance is closely tied to the effectiveness of SAM. If SAM encounters difficulties, such as producing inaccurate or poor segmentations, it can adversely affect the performance of the tracking framework. The quality of the object mask prediction by SAM directly impacts the tracking accuracy and robustness. Therefore, in scenarios where SAM struggles or fails to provide precise segmentations, the tracking performance may indeed experience a significant drop, highlighting the method's dependency on SAM's success in providing accurate object masks.

**Questions:**

Can you show some failure cases of the proposed method? Is that related to the SAM results?

**Details Of Ethics Concerns:**

No concerns.

---

### Official Review · Reviewer_tNTu · 2023-11-01

**Soundness:** 2 fair
**Presentation:** 2 fair
**Contribution:** 2 fair
**Rating:** 3
**Confidence:** 4

**Summary:**

The core idea of this paper is that points on the same object are physically bound and should share the same motion statistics. To achieve this, the authors propose to average the initial motion estimates of some points on the object as the motion of the whole object. Then crop the video frames along the object trajectory to achieve high precision point trajectory tracking.

**Strengths:**

1. The proposed scheme is easy and effective, and its improved performance is demonstrated on various datasets as well as on downstream tasks.

**Weaknesses:**

1. Eq 5 shows the present method takes average pixel displacements as the object motion and claims that it is a reliable instance-level motion estimation. But directly calculating the average pixel displacements is valid if and only if the object only translates in the image plane. This cannot be valid if the object is rigid and there is affine motion such as rotation, or if the object is non-rigid. I agree that such a naive assumption can be used as an initialization for model optimization, but I don't think it's an exciting innovation to elaborate on such great length.
2. the model Seg, which produces the segmentation mask for the initial frame. This method uses SAM for preprocessing, is it only for the first image or for all video frames? How do the later frames establish associations with the points in the earlier frames? This procedure needs further clarification.
3. InstaTAP can be built on top of any existing point tracker. But there is no discussion in experiments.
4. Lack of computational complexity analysis. Looks like a very heavy optimization process.

**Questions:**

1. Does the proposed mechanisms require to retrain the existing point tracking models? Or just need to use publicly available pre-trained models?

---

### Author Response · Authors · 2023-11-18
**Author Rebuttal by Authors**

Dear reviewers and AC,

We sincerely appreciate your valuable time and effort spent reviewing our manuscript.

As reviewers highlighted, we believe our paper is generally well-written (sKxk), and the proposed scheme is easy and effective (tNTu), with the interesting idea for motion prediction (K1PM), demonstrated to be effective in the experiments (tNTu,sKxk).

We appreciate your constructive comments on our manuscript. In response to the comments, we would like to clarify several important concerns and questions:

1. The assumption behind Eq. 5 cannot be valid (tNTu).
First, we would like to point out Eq. 5 is only responsible for approximating the instance-level motion rather than the final point tracking.

In fact, eq. 5. is intended to average out rotations, which do not significantly translate the mask region occupied by an instance. We note that rotations and non-rigid motions are handled by the subsequent high-fidelity tracking stage (Sec 3.4), where we do not average multiple points.

2. The contribution of this framework is limited (K1PM, sKxk).
Our contribution is not limited to simply using SAM. For example, one could stack SAM for multiple video frames to estimate the instance motion, but it suffers from severe noises and huge computation costs for segmenting multiple frames. Instead, our method relies on SAM for only once at the first frame, and then tracks the semantic neighbor points through our sophisticated procedure Eqs. 2-7, in an efficent manner.

Moreover, our performance boost is not solely due to the instance motion powered by SAM. For instance, our proposed high-fidelity point tracking enhances the high-frequency details in the visual feature (Sec. 3.4), and boosts the performance significantly. In Tab. 3, we provide the ablation study, finding a sole contribution by the module (Clip), e.g., 28.3 -> 35.4 J-1 metric.

Thank you very much!

Authors.